Exploring the density and morphology of coconut structures at two locations: a time-based analysis using computer tomography

Lin Shenghuang 1
Sun Chengxu 2
Luo Li’an 3
Huang Mengxing 4
John Martin JeromeJeyakumar 2
Cao Hongxing 2
Hu Jinyue 1
Bai Zhiming 1
He Zhanping 1
Zhang Yu yuzhang2015@hainanu.edu.cn 5
Chen Jing jingchen_haiko@163.com 1
1 Central South University Xiangya School of Medicine Affiliated Haikou Hospital , Haikuo , China
2 Coconut Research Institute, Chinese Academy of Tropical Agricultural Sciences , Hainan , China
3 Siemens Healthineers , Guangzhou , China
4 College of Information and Communication Engineering, Hainan University , Hainan , China
5 College of Computer Science and Technology, Hainan University, Haikou , Hainan , China
Marunaka Yoshinori
Electronic publication date: 2024 Oct 14
Publication date: 2024
Volume: 12
Electronic Location ID: e18206
Received 2024 Apr 29; Accepted 2024 Sep 10
Copyright: ©2024 Lin et al.
Copyright year: 2024
Copyright holder: Lin et al.
License: This is an open access article distributed under the terms of the Creative Commons Attribution License, which permits unrestricted use, distribution, reproduction and adaptation in any medium and for any purpose provided that it is properly attributed. For attribution, the original author(s), title, publication source (PeerJ) and either DOI or URL of the article must be cited.
License URL: https://creativecommons.org/licenses/by/4.0/

Keywords: Coconut, Computer tomography, Postprocessing technique, Non-invasive method

Funding: Central Finance Forestry Science and Technology Promotion Demonstration Fund Project Qiong [2021] TG 05 Key R&D Project of Hainan Provincial Department of Science and Technology ZDYF2019040 This study was supported by the Central Finance Forestry Science and Technology Promotion Demonstration Fund Project (Qiong [2021] TG 05) and the Key R&D Project of Hainan Provincial Department of Science and Technology (ZDYF2019040). The funders had no role in study design, data collection and analysis, decision to publish, or preparation of the manuscript.

==============================
Background

The study aimed to observe the internal structure of coconuts from two locations (coastal and non-coastal) using computed tomography (CT).

Methods

Seventy-six mature coconuts were collected from Wenchang and Ding’an cities in Hainan Province. These coconuts were scanned four times using CT, with a two-week interval between each scan. CT data were post-processed to reconstruct two-dimensional slices and three-dimensional models. The density and morphological parameters of coconut structures were measured, and the differences in these characteristics between the two groups and the changes over time were analyzed.

Results

Time and location had interactive effects on CT values of embryos, solid endosperms and mesocarps, morphological information such as major axis of coconut, thickness of mesocarp, volume of coconut water and height of bud (p < 0.05).

Conclusions

Planting location and observation time can affect the density and morphology of some coconut structures.

Introduction

Coconut (Cocos nucifera L.) is a perennial palm tree that is widely cultivated on tropical islands and coastlines. Coconut is an important food raw material in the tropics, with rich nutrition (DebMandal & Mandal, 2011). There are differences in the taste and nutritional composition of coconut in different regions. Different regions, such as coastal city and non- coastal city, may have different impact on the growth of coconut due to their differences in climate, soil and water. The study on the differences in coconut structure in different regions can provide some information on the regional influence of coconut breeding.

Previous studies have utilized various methods examined solvent extracts or the chemical composition of the coconut plant for biotechnological or biomedical research purposes (Roopan, 2016; Narayanankutty et al., 2022; Zhang et al., 2022). However, these methods require cutting coconuts for observation which cannot be analyzed non-invasively. In fact, macroscopic observation of coconuts holds immense value, as it allows for the analyze the impact of agricultural management and climate conditions on coconut planting (Kumar et al., 2008; Hebbar et al., 2022). Few other studies focused on coconut morphology and fruit quality traits have been analyzed for germplasm resource evaluation and identification (Zhang et al., 2021). By evaluating the morphology and fruit quality traits, researchers can identify and classify the germplasm resources available. This information is crucial for conservation efforts and for selecting the right varieties for specific purposes, such as commercial cultivation or breeding programs. Understanding the internal morphological structure of coconuts is challenging without invasive observation tools. This is primarily due to the thick mesocarp and hard endocarp layers. These layers act as barriers, making it difficult to visually examine the internal structure of the coconut without causing damage. Consequently, researchers and scientists face obstacles in studying the intricate arrangement of tissues and components within the coconut. Overcoming these challenges requires the development of non-invasive techniques that can provide detailed insights into the internal morphology of coconuts.

Using medical computer tomography (CT) scanning to observe coconuts without harm is a groundbreaking study that allows researchers to measure the morphological information of the internal structure without causing any invasiveness. CT computes the attenuation of X-rays as they pass through the object, CT can construct highly accurate images that show the spatial distribution of X-ray attenuation (El-Khoury, Bennett & Ondr, 2004). Each voxel value in a CT data represents the X-ray attenuation at that position, known as CT value. The CT value mainly reflects information on material density, whose unit is Hounsfield units (HU). After processing, CT can obtain two-dimensional (2D) slice and three-dimensional (3D) model. Traditional CT 2D images are gray-scale two-dimensional cross-sectional images, the larger the CT value in the 2D image, the higher the material density in the location, and vice versa. The 3D model uses different colors and transparences to model the object based on the difference in CT values, showing the internal structure of the object in a more realistic way (Lin et al., 2023). In the medical field, CT is commonly used to observe and diagnose the anatomical morphology and diseases of human organs such as the bones, heart, and blood vessels, such as to diagnose the location of fractures or blood vessel bleeding (Gou et al., 2021; Moroni et al., 2022; Di Paolo et al., 2020). It overcomes the overlap between tissues or organs and facilitates the detection of lesions, an advantage that can also be extended to more application scenarios.

In agricultural applications, X-ray computed tomography can be used for quality inspection of agricultural products, such as internal quality evaluation, defects and contaminants detection, microstructure observation, mechanical property measurement, and others (Haff & Toyofuku, 2008; Du et al., 2019). In addition, some researchers explored plant internal morphology using X-ray technology, for example, modeling root morphology in soil (Metzner et al., 2015; Teramoto et al., 2020; Subramanian et al., 2015), mapping the vascular structure of cucumbers (Sui et al., 2021) spatio-temporal evaluation of waterlogging-induced aerenchyma formation in barley (Kehoe et al., 2022). Some researchers have gone further by using X-rays to analyze differences in the internal structure of different varieties of plants to gain insight into plant development, such as identifying possible biophysical constraints during development through allometric relationships in nutshell (Amézquita et al., 2024) or observing the Witch’s Broom bud sport of grapevine development from buds to shoots (Ritter et al., 2024). However, at present, no studies have been conducted to analyze the differences in the internal structure of different types of coconuts based on X-ray. Our previous studies used CT imaging of coconuts to establish the feasibility of visualization and quantitative analysis of the internal structure of coconuts with CT data (Lin et al., 2023).

Using medical CT, this study analyzed the internal structure of coconuts of the same variety at maturity stage in two different areas (coastal and non-coastal), and explored the regional differences in the internal morphological structure and density of coconuts.

Materials and Methods

This study was approved by the institutional review board of Haikou Affiliated Hospital of Central South University Xiangya School of Medicine. The coconut samples were obtained with the approval of the Coconut Research Institute.

The general materials

About 76 coconuts of the same variety between the ages of 10–12 months came from two different cities in Hainan, Ding ’an and Wenchang, of which Ding’an is non-coastal city, and Wenchang is a coastal city. The coconuts appeared intact and undamaged (Fig. 1). Forty-eight coconuts from Ding’an were from coconut trees that were approximately 20 years old, with a height of over 10 m and around 32 leaves. They are 10–12 months old coconuts as judged by coconut farmers and grew in sandy soil. On the other hand, twenty-eight coconuts from Wenchang came from coconut trees that were approximately 15 years old, with a height of over 14 m and around 25 leaves (Fig. S1). The coconuts, grown in the red sandy soil, were assessed by the coconut farmers to be approximately 10 to 12 months old.

Figure 1 Photos of coconuts from Ding ’an and Wenchang before scanning.

The coconuts from both locations underwent four CT scans at two-week intervals. Prior to each scan, the weight of each coconut was measured and recorded. In order to conduct a thorough examination, the coconuts were carefully positioned on a porous plastic film that was placed on the examination bed. Twelve coconuts were placed on a mold for scanning arranged in two columns with six rows.

Image acquisition

A dual-source CT scanner (SOMATOM Definition Flash; Siemens Healthineers, Forchheim, Germany) was used for all examinations. Four CT scans were completed at four time points (Time 1–4) on September 19, October 03, October 17, and October 31, 2021, with a total duration of 43 days. Each time, the coconut stem end faced upwards while its top was positioned on fixed film holders to ensure consistency throughout subsequent scans. The scanning parameters remained consistent: slice thickness/increment = 0.6 mm/75%; tube voltage = 120 kv; tube current = 250mAs; field-of-view (FOV) = 400 mm × 400 mm; rotation speed = 0.5s/r. Voxel size = 3.125*3.125*0.6 mm3; Scanning time =18–23 s; Radiation dose: Dose Length Product (DLP) = 695-778.5 mGy*cm.

Image processing

The CT raw images were collected and transferred to the Siemens Syngo. via post-processing workstation. The 2D images displayed sectional views of coconut from various angles, revealing the internal structure’s two-dimensional section morphology and spatial relationship. Parameters such as CT values, thicknesses, lengths, and volumes of each structure could be measured on 2D images. Measurement was done manually, averaging the values of three ROIs measured on each coconut at several slices to reduce measurement errors. The 3D model image generated by post-processing could show the three-dimensional morphology and spatial position relationship of each structure of coconut. Different 3D model templates can visualize the 3D spatial relationship of different structures of coconut by designing the transparency and color of voxels with different densities in CT data (as shown in Fig. 2).

Figure 2 Illustration of coconut structure in CT images.

Images from left to right are longitudinal 2D images and 3D models under three different templates, respectively.

The study conducted by the Coconut Institute researchers involved the transmission of images for measurement. The CT values of various parts of the coconut, including the embryo, bud, solid endosperm, coconut water, mesocarp, endocarp, and coconut apple (Table S1). Coconut weight was measured using a weight scale. The morphological parameters of coconut, such as major and minor axes of whole coconut, volume of coconut water (also known as liquid endosperm), solid endosperm thickness, endocarp and mesocarp thickness, major and minor axes of coconut apple, bud height, major and minor axes of coconut water, were measured in different sections of 2D images. The study did not measure the exocarp thickness or the size of some coconut apples due to the thinness of the exocarp and the small size of those apples. The morphological parameters of a coconut can be defined by measuring its major and minor axes. The major axis refers to the diameter in the upper and lower directions of the coconut’s longitudinal section, while the minor axis refers to the diameter in the left and right directions. The major and minor axes of the liquid plane of coconut water serve as important indicators of its height and width, respectively. As illustrated in Fig. 3, the major axis represents the vertical dimension of the liquid plane, providing valuable information about the overall level or depth of the coconut water. On the other hand, the minor axis reflects the horizontal dimension, giving insights into the width or breadth of the liquid surface, At the same time, the internal structural characteristics of the coconut were also revealed by CT scans (Fig. S2).

Figure 3 Definition of morphological parameters during measurement.

The major axis represents the vertical dimension of the liquid plane, providing valuable information about the overall level or depth of the coconut water. The minor axis reflects the horizontal dimension, giving insights into the width or breadth of the liquid surface.

Statistical analysis

All statistical analyses were performed using SPSS (version 25, IBM Corp), and MATLAB (R2021b) was utilized for graph plotting. Quantitative data were presented as mean ± standard deviation. The mean and standard deviation of the scanned data at each time point were calculated, plotted, and repeated. An analysis of variance (ANOVA) was employed to calculate the difference between CT value and the morphological parameters of coconut at four time points. The measured parameters were analyzed for differences between two locations at four time points using the between-subject effects followed by the within-subject effects to examine the interaction effect between time and location on coconuts. The intraclass correlation coefficient (ICC) was used to assess consistency in CT data from a specific variety that underwent three repeated measurements.

Results

The 2D images and 3D models obtained by CT offer a comprehensive visualization of the internal structure of coconut. Statistical results showed that both time and location had interactive effects on CT values of embryos, solid endosperms and mesocarps as well as weight, major axis of coconut, thickness of mesocarp, volume of coconut water and height of bud (p < 0.05).

Subjective analysis

The CT images of a coconut from Ding’an captured in four scans over two-week intervals are shown in Figs. 4 and 5, while the CT images of a coconut from Wenchang are depicted in Figs. 6 and 7. The images obtained through CT scans provide a detailed insight into the internal structural morphology and spatial relationships of coconuts. These scans allow for the observation of changes in the coconut’s composition over the course of four scans. The CT images of a coconut reveal its internal structure, showcasing the coconut apple, root, bud, and the spatial relationship between the exocarp, endocarp, coconut apple, and coconut water. In the two coconuts shown in Figs. 4–7, at the initial scanning time point, small coconut apples were detected in the inner endocarp (the inner cavity of the coconut). Additionally, buds were observed in the mesocarp, accompanied by a limited number of small roots around the buds. These early developmental stages provide valuable insights into the growth and maturation process of coconuts. The continuous CT images display the gradual growth of coconut apples and buds over time. Buds sprout from mesocarp to outer pericarp, while roots increase, thicken, and lengthen, with diverse morphology. Despite the gradual reduction in coconut water, the volume and size of coconuts have not changed significantly. This indicates that the fruit’s structure remains largely consistent despite alterations in its liquid content. The ability of coconuts to maintain their size and shape while losing water suggests a resilient and stable composition, making them a reliable source of refreshment and nutrition.

Figure 4 The 2D images and 3D models obtained from four CT scans of coconuts from Ding ’an.

From left to right, the images at four time points with an interval of two weeks were shown.

Figure 5 The 3D models of four CT scans of a Ding’ an coconut.

The four rows from top to bottom are the images of four time points at the interval of two weeks, and the images in each row are 3D model images from different angles at one of the time points. 3D images showed the morphology of budding and root and their spatial relationship with endocarp.

Figure 6 The 2D images and 3D models of Wenchang coconut from four scans.

Figure 7 The 3D models from four scans of Wenchang coconut.

Two rows of images show the morphological changes of shoots and roots at four time points from two perspectives, respectively.

Objective analysis

The mean CT value and standard deviation of each structure of coconut were calculated and plotted (Fig. 8). Details are as follows:

(1) Embryo: the CT value of coconut in both sites decreased slowly with time, and the CT value of coconut in Ding’an was slightly lower than that in Wenchang. The value of the Ding’an coconut decreased from −425 ± 284 HU (time 1) to −750 ± 100 HU (time 4), and that of Wenchang decreased from −338 HU ± 297 HU (time 1) to −725 ± 117 HU (time 4).

(2) Bud: The CT values ranging from −100 HU to 100HU, with a majority falling between −50 HU and 50 HU show consistent distribution. The CT values being unrelated to location and time.

(3) Solid endosperm: The CT value falls between 0 and 50 HU, it indicates a specific range of tissue density. The lack of a significant difference between two locations within this range suggests similar tissue composition or characteristics.

Figure 8 (A–F) CT values of each structure of coconut at the two locations.

Time 1–4 are the four time points observed consecutively in four scans at two-week intervals.

Endocarp: The CT values of coconut from Ding’an and Wenchang were normally distributed, with 196 ± 47 HU and 184 ± 49 HU, respectively.

(5) Mesocarp: CT values were distributed between −950 and −750 HU.

(6) Coconut water: CT value was normal distribution, 13 ± 5 HU for Ding’an coconut and 17 ± 6 HU for Wenchang coconut.

Differences between locations were analyzed for each time point, and the results showed significant differences (p < 0.05) only at certain time points for the embryo and mesocarp, as well as three time points for coconut water. Other parameters had no significant effects due to location factors. The average values of Ding’an and Wenchang coconuts at four different time points (between-subject effects in Table 1) showed that only the CT value of coconut water had significant differences (p < 0.05) between the two groups from two locations; other parameters did not show significant differences. According to the ANOVA (within-subject effects in Table 1) showed that the CT values of embryo, bud, solid endosperm, endocarp, mesocarp, and coconut water all had significant temporal main effects (p < 0.05). Results from cross-effects analysis between time and location (time × location) indicated that there was an interaction effect on CT value between both factors for embryo, endosperm, and mesocarp (p < 0.05).

Table 1 P value of repeated measures analysis of variance (ANOVA) results of CT values of coconut structure from two locations.

	Embryo	Bud	Solid endosperm	Endocarp	Mesocarp	Coconut water	
Time1	0.284	0.67	0.051	0.081	0.71	0	
Time2	0.002	0.224	0.24	0.08	0.212	0.015	
Time3	0.215	0.39	0.284	0.141	0.007	0.095	
Time4	0.443	0.108	0.896	0.273	0.569	0.004	
Between-subject effects	
Location	0.126	0.371	0.966	0.101	0.548	0	
Within-subject effects	
Time	0	0.002	0.002	0.024	0.002	0.005	
Time x Location	0.05	0.367	0.028	0.062	0.004	0.268	

The morphological parameter measurements of the coconut were calculated and plotted (Fig. 9). Details are as follows

(1) Weight: the values decreased with time, from 1,050 ± 200 g to 903 ± 141 g of Ding’an coconut; Wenchang coconuts decreased from 1,002 ± 185 g to 940 ± 185 g, and the location differences between the four observation time points were significant.

(2) Major axis of coconut: the average value of coconuts in Ding’an was slightly larger than that in Wenchang, but both values were between 16–22 cm.

(3) Minor axis of coconut: the data showed a normal distribution. The average value of coconut in Ding’an was slightly smaller than that of Wenchang (15.90 ± 1.00 cm vs. 16.37 ± 0.47 cm).

(4) Major axis of coconut apple: the coconuts in Ding’an were larger than that in Wenchang, and the value of both increased steadily, from 4.09 ± 1.96 cm to 6.64 ± 2.26 cm in Ding’an, and from 2.77 ± 1.67 cm to 5.39 ± 2.27 cm in Wenchang.

(5) Minor axis of coconut apple: the coconuts in Ding’an were larger than that in Wenchang, and the value of both increased steadily, from 3.66 ± 2.11 cm to 6.30 ± 2.01 cm in Ding’an, and from 2.35 ± 1.38 cm to 4.70 ± 1.97 cm in Wenchang.

(6) Endocarp thickness: the values were between 0.1−0.4 cm.

(7) Mesocarp thickness: the data showed a normal distribution, 2.31 ± 0.45 cm for Ding’an coconut and 2.19 ± 0.46 cm for Wenchang coconut.

(8) Major axis of coconut water: the data in Ding’an coconut were smaller than that in Wenchang, with 7.80 ± 0.85 cm in Ding’an and 8.42 ± 1.01 cm in Wenchang.

(9) Minor axis of coconut water: the data in both locations decreased with time, from 5.39 ± 1.38 cm to 4.08 ± 1.38 cm in Ding’an; Wenchang decreased from 5.13 ± 1.61 cm to 4.15 ± 1.26 cm, and the values of Ding’an and Wenchang changed extremely significantly with time.

(10) Thickness of solid endosperm: the values of coconuts in Ding’an were lower than that in Wenchang, with a normal distribution of 1.02 ± 0.15 cm for coconut in Ding’an and 1.08 ± 0.19 cm for Wenchang.

(11) Volume of coconut water: The value of Ding’an coconuts decreased from 159.35 ± 51.45 cm3 (time 1) to 152.34 ± 79.67 cm3 (time 4), and Wenchang from 225.62 ± 101.45 cm3 (time 1) to 153.49 ± 108.78 cm3 (time 4). The changes of Ding’an and Wenchang with time were extremely significant.

(12) The height of bud: the value of coconut in Ding’an was greater than that in Wenchang, and both showed an upward trend, from 3.25 ± 2.18 cm to 7.28 ± 5.03 cm in Ding’an, and from 1.70 ± 0.76 cm to 3.12 ± 1.54 cm in Wenchang. The value in Ding’an and Wenchang changed extremely significantly with time.

Figure 9 Morphological parameters of coconut in the two locations.

Time 1–4 are four time points at two-week intervals.

The study analyzed location differences at four time points and found significant variations in major axis of coconut, major axis of coconut water, and height of bud consistently across all four time points. These findings suggest that location plays a crucial role in determining the physical characteristics of coconuts and their growth stages. Significant variations were observed in the sizes of the coconuts’ minor axis, as well as the major and minor axis of coconut apple. Additionally, variations were found in the thicknesses of the mesocarp and solid endosperm and in the volume of coconut water at specific time points. The study found that no significant differences were observed between locations for the remaining features at each time point. The average values of major axis of coconut, major and minor axis of coconut apple, major axis of coconut water, and height of bud between Ding’an and Wenchang coconuts showed extremely significant differences (p < 0.01) according to Table 2 (between-subject effects). The minor axis of coconut and thickness of solid endosperm exhibited significant differences (p < 0.05) between the coconuts from two locations.

Table 2 P value of repeated measures analysis of variance (ANOVA) results of morphological parameters of coconut from two locations.

	Weight	Major axis of coconut	Minor axis of coconut	Major axis of coconut “apple”	Minor axis of coconut “apple”	Thickness of endocarp	Thickness of mesocarp	Major axis of coconut water	Minor axis of coconut water	Thickness of solid endosperm	Volume of coconut water	Height of bud	
Time1	0.309	0	0.104	0.179	0.178	0.157	0.237	0.046	0.492	0.418	0.001	0.01	
Time2	0.991	0	0.084	0.025	0.018	0.507	0.47	0.003	0.824	0.13	0.643	0.002	
Time3	0.296	0.008	0.022	0.042	0.053	0.058	0.011	0.005	0.421	0.026	0.244	0.009	
Time4	0.146	0	0.014	0.052	0.006	0.873	0.719	0.014	0.826	0.79	0.959	0.002	
Between-subject effects	
Location	0.605	0	0.037	0.009	0.009	0.273	0.139	0.001	0.914	0.036	0.09	0.002	
Within-subject effects	
Time	0	0	0.132	0	0	0	0.408	0.013	0	0.001	0	0	
Time x Location	0.002	0	0.182	0.245	0.211	0.363	0.001	0.665	0.186	0.583	0	0.006	

The rest of the features did not show any significant differences. The ANOVA (see within-subject effects in Table 2) revealed that except for minor axis of coconut and mesocarp thickness which had no significant time effect, all other features had a significant time main effect (p < 0.05). Results from the interaction analysis between time and location (time × location) indicated that both factors had an interactive effect on weight, major axis of coconuts, mesocarp thicknesses, volume of coconut water, and height of bud (p < 0.05).

Discussion

In this study, we can delve into the internal structure of a coconut, and observe properties of coconut apples, the changes of coconut water, and the process of buds growing from the mesocarp to the outside of the exocarp. Statistical results showed that both time and location had interactive effects on CT values of embryos, solid endosperms and mesocarps, morphological information such as major axis of coconut, thickness of mesocarp, volume of coconut water and height of bud (p < 0.05). CT value of coconut embryos decreased significantly with time. which may relate to the growth and development of coconut. The interactive effects of time and location on CT values and morphological information may suggest that these structures are influenced by planting area and observation time.

Previously, many research methods were used to analyze coconut in different dimensions, such as using Raman spectroscopy and stoichiometry to quantify the sugar content of coconut water (Richardson et al., 2019), using machine learning algorithms based on acoustic signals to determine the maturity of coconut (Caladcad et al., 2020). In this study, medical CT was used to observe the whole coconut non-invasively, and to explore the differences in structural density and morphological properties of the same species of coconut in different locations and changes over time. Although the X-ray imaging used in CT scans has radiation, the radiation dose of medical CT is low, and the widespread use of X-rays in agriculture also indicates that radiation effects on plants are acceptable (Haff & Toyofuku, 2008; Du et al., 2019). In addition, magnetic resonance imaging (MRI) in medical imaging is also a technology that can realize non-invasive overall observation, and this technology is also applied in agriculture (Mariette et al., 2012). However, MRI scanning time is too long and special coils are required, and CT is more convenient, so the latter imaging is adopted in this study.

Our study has some limitations. First, the observation of coconut density and morphological changes has not been further analyzed with botanical information, failing to provide a more effective reference for botanical research, which needs to be further explored by subsequent studies; second, a single variety of coconut was used in this experiment, and more information related to varieties is worth further classification and exploration; third, we only use CT to analyze the coconut structure, and more information can be used to analyze the coconut, such as chemical examination of the nutritional content of the coconut, and microscopic imaging to understand the subtle differences in the coconut structure.

Conclusion

The study demonstrates the potential of using CT for analyzing coconuts, including observing morphological and density characteristics. Planting location and observation time can affect the density and morphology of some coconut structures.

Supplemental Information

Supplemental Information 1 Test materials, test images, and data record for coconut CT testing

The CT values of various parts of the coconut, including the embryo, bud, solid endosperm, coconut water, mesocarp, endocarp, and coconut apple.

Supplemental Information 2 CT values and Morphology parameters of coconut

The test materials and data record tables for coconut CT testing

Additional Information and Declarations

Competing Interests

Author Contributions

Data Availability

Li’an Luo is an employee of Siemens Healthineers, but she was not involved in the analysis of the data and had no control of data or information submitted for publication. There is no conflict of interest or industry support in this study. The other authors declare that they have no competing interests.

Shenghuang Lin conceived and designed the experiments, performed the experiments, prepared figures and/or tables, authored or reviewed drafts of the article, and approved the final draft.

Chengxu Sun conceived and designed the experiments, performed the experiments, prepared figures and/or tables, authored or reviewed drafts of the article, and approved the final draft.

Li’an Luo performed the experiments, authored or reviewed drafts of the article, and approved the final draft.

Mengxing Huang performed the experiments, authored or reviewed drafts of the article, and approved the final draft.

JeromeJeyakumar John Martin performed the experiments, authored or reviewed drafts of the article, and approved the final draft.

Hongxing Cao conceived and designed the experiments, analyzed the data, prepared figures and/or tables, organized and supervised the overall project, and approved the final draft.

Jinyue Hu conceived and designed the experiments, performed the experiments, prepared figures and/or tables, and approved the final draft.

Zhiming Bai conceived and designed the experiments, analyzed the data, prepared figures and/or tables, authored or reviewed drafts of the article, and approved the final draft.

Zhanping He conceived and designed the experiments, analyzed the data, prepared figures and/or tables, authored or reviewed drafts of the article, and approved the final draft.

Yu Zhang conceived and designed the experiments, analyzed the data, prepared figures and/or tables, authored or reviewed drafts of the article, and approved the final draft.

Jing Chen conceived and designed the experiments, analyzed the data, prepared figures and/or tables, authored or reviewed drafts of the article, and approved the final draft.

The following information was supplied regarding data availability:

The raw data are available in the Supplementary Files.

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
