# Peer review of "Exploring the density and morphology of coconut structures at two locations: a time-based analysis using computer tomography"

_PeerJ, doi:10.7717/peerj.18206_

## Round 0.1 · original submission · Major Revisions

In order to expedite the decision to the authors, Staff are progressing this manuscript at this time. Please respond to all the peer-review comments in detail

·

Basic reporting

- The abstract does not include any result or conclusion from the results. I found it too generally described.
- The introduction mentions earlier studies on growth using X-ray technology only very briefly (line 84). I would like to see this to be extended.
- X-ray imaging is suggested, but not compared to other imaging technologies such as MRI. Could that be included as well?
- The relevance of the study (line 90-96) is quite generally described. As I understand, the researchers focused on following the growth of the coconut, so I do not see a direct relationship with medicinal composition and bioactive compounds (line 95).
- Results are given for time 1 until 4, but I cannot find this back in the methodology (or I overread it?)? What timings do these mean? Is it the two-week intervals? Better to always use the same term, or to explain it in the caption of your figures.
- Discussion: line 352: ‘effect of X-rays on the coconuts’ – Do you expect that X-rays are harmful to coconuts? Some studies claim that X-rays are safe for food (at a fair energy level though).
- Conclusion: I found it too general. Line 369: ‘Results suggest that both time and location have interactive effects on the CT values and physical properties of coconuts.’ Could you elaborate more on those specific results, what to conclude from that?

Experimental design

- Introduction: there was a lot of focus on the nutritional characteristics. However, I believe that X-ray CT can rather contribute to food processing. Can that aspect be considered?
- Research question: Remark related to the one above: Why can it be relevant to gain insight into the growth process of the coconuts? Too generally defined now.
- Research question: Could you elaborate on why it is relevant to take two locations into account for the same cultivar?

Validity of the findings

- Table 1 + 2: Meaning of ‘df’ is not included in the caption? Degrees of freedom? Is P, the p-value from the statistics? I doubt if it is even meaningful to include your degrees of freedom here.
- Nice figures from the CT data! T

Additional comments

no comment

·

Basic reporting

1. With the exception of a few typos, the article is in professional English. There are a few instances where the use of strictly medical CT terminology limits the audience, since many who use CT in research use industrial CTs or synchrotron facilities, which do not follow medical CT conventions. Additionally, the statistical methods description should be in the methods section, not the results.
2. There is a distinct lack of relevant literature. While CT has not been as widely adopted in plant studies as compared to animal studies, it is not as small as the authors show in this article. The literature in the manuscript mainly focuses on genetic studies and/or other methods for looking at coconuts, but little time is spent connecting them to the current study and one sentence is spent on CT use in plant biology/agriculture, which ignores a wealth of papers from several different groups that use CT extensively that would help support their study's rationale and the discussion. I am not sure if this is a product of the authors limiting their literature search to medical CT related literature or if there is some other reason for this. Food science is another discipline that the authors may find relevant literature from as CT has been studied to monitor disease and bruising, for example.
3. While the authors have well organized tables and figures and present some raw data, I am disappointed that the CT scans themselves are not available. While CT scans can be harder to upload as they can be rather large, third-party sites such as MorphoSource or Dryad should be able to handle the data. Medical CT scans are much smaller (KB to MB range) than Industrial CT scans (GB range) and numerous groups using Industrial CT have posted their data to these and similar sites.
4. There is a lack of a clear hypothesis and/or reasoning behind this study. The authors state in their guidance that they are validating another study and bridging knowledge gaps, but in the paper none of that is noted, discussed, nor presented in anything more than general terms and the validation aspect is not discussed at all. It makes it very hard to understand why their research was done and what it adds. CT has been used as a non-destructive method since the 1980s in several fields and its use as a non-destructive method is well known. The authors should focus on why that is relevant to coconuts. The authors make several general statements like how it can look at morphology, but there is no further presentation of why this is important. In the introduction, the discussion of molecular and genetic measurements are not connected to how they relate to the morphology analyzed through CT scans, for example. Part of this is the lack of relevant CT literature, which would give support to why CT is relevant to coconuts based on studies from other plants/agricultural products. The introduction leaves me unsure what knowledge gap the study is filling. An example of this is the authors discuss medical compound research as an avenue for which CT could be helpful. However, those compounds would not be visible on CT scans, so it is difficult to connect how CT scans would assist in that research. Furthermore, it does not explain how the study will show CT’s usefulness to that avenue of coconut research.

Experimental design

1. This article fits within the aims and scope of this journal.
2. The research question does not exist in this article other than in general terms. The authors present a reasonably well thought out experiment that I can think of several reasons it would be relevant, but it is not stated well, if at all, in the paper. It feels like a good method and results with a random introduction and discussion added. While the study looked at the density and structure of the coconuts, no reasoning for why this was done is ever stated and the discussion barely even mentions the results and does not connect it to any broader literature. The authors suggest this is a proof of concept, but there is no rationale for why they studied what they studied. If the authors wanted to show this as a validation of a previous study, as suggested in the reviewer notes, then it should be explained.
3. The research itself follows the norms and standards of this type of research with a few minor exceptions, specifically standard deviation used instead of standard error.
4. The methods section is very good with the exception of not listing the voxel size, thresholding methods should be more thoroughly explained, and exposure time of the images should be noted.

Validity of the findings

1. The authors present this in the guidance as a replication study of a previous study, but give no indication of this in the paper and no support for why this study even took place.
2. The lack of raw CT data makes it hard to assess the validity of the results. The measurements themselves are presented, but without the CT scans, the validity of those measurements cannot be assessed.
3. The conclusion is supported by the data, but has little connection to the discussion. If the points raised in the conclusion were discussed in the discussion, this would be a much better paper.

Additional comments

This paper has potential, but the lack of rationale and discussion greatly deprives the paper of relevance. The authors designed a nice, well thought out study and then provide almost no reasoning for it and no discussion of the results and how it addresses knowledge gaps and/or adds to the literature. The conclusion has more relevant discussion than the discussion. Most of the introduction is disconnected from the study. The authors give background on why coconuts are important, then discuss genetic and metabolic studies, but do not connect how these relate to CT, since while morphological things can be measured with CT, genetic and metabolic things cannot. There is no connection between why the authors are discussing the genetic and metabolic studies and why CT is a relevant method. This lack of connection and flow make the introduction very difficult to read and understand the reasoning behind the author’s study. The conclusion and guidance to the reviewer presents more rationale for the study than the introduction.

Specific comments:
Line 75: Pixel should be voxel. The basic element of a CT scan is a volume element (voxel) not a picture element (pixel). Pixel is 2D, while voxel is 3D. Even if you are looking at a single 2D slice, the greyscale value is based on the volume represented by that slice.

Line 76-78: The CT value should be noted that they are Hounsfield units. While Hounsfield units are standard in medical CTs, someone from outside the medical field may not know this. It is also arguable how relevant/applicable Hounsfield units are to coconuts as HUs are very specifically calibrated to human tissue.

Line 77-78: MPR is terminology specifically used in medical CT. This is used because CT images of the human body are taken in a certain orientation and the MPR designation indicates that they are looking at a different orientation from the standard scan direction. However, scanning of coconuts does not have a standard orientation. Using the term 2D slices would be more appropriate in this case. I understand the author’s are mimicking the terminology used in Lin 2023, but the medical terminology is very limiting especially if researchers reading this are using synchrotron or industrial CTs where this terminology does not exist.

Line 83-84: This is missing a large amount of research from agriculture with CT. Root morphology has been extensively looked at, beyond the three references mentioned here (Chris Topp’s group at Danforth Center and the Hounsfield facility at the University of Nottingham, for example). Dan Chitwood’s group at Michigan State University has looked at barley, citrus, and walnuts using CT. I know there are others, including flower morphology studies and the food science literature has studied uses of CT for disease and bruising monitoring.

Line 125-Line 130: What was the exposure time of the images, i.e. how long each image took to capture? Another way to state this would be the amount of time the total scan took. Medical X-rays are usually limited on their exposure time, but it is something that can be adjusted, so it is important to note. Additionally, were phantoms used to calibrate the CT values or was it an internal setting within the software? Additionally, what was the voxel size of the images? The voxel size limits the accuracy of the measurements taken by the researchers, since you cannot measure smaller than the voxel size.

Line 151-154: How were these morphological traits measured? Was it measured from one slice or was it measured on several and then averaged? Was it measured by hand or was an algorithm used? How this was accomplished can greatly affect repeatability.

Line 215-225: Usually, the statistical analyses explanations are included in the methods section, not the results section. Please move them.

Line 219: When using an ANOVA it is customary to use the standard error, instead of the standard deviation.
Line 313: ANOVAR? Typo?

Discussion is weak to non-existent. The first limitation of the study is probably not a limitation. Radiation studies on plants are well documented, but are mostly done using radioactive elements. However, these studies are still looking at ionizing radiation dosages. The United Nations Scientific Committee on the Effect of Atomic Radiation showed in 1996 there is no detrimental effect if the dose does not exceed 10 mGy/d for chronic dosage, which is less than the 5 mGy/d recommendation for humans. As the instrument used was a medical CT designed for safe use in humans, it is highly likely plants, with their higher radiation tolerance, were unaffected by the scanning.
The authors state in the review notes that this in a validation of an earlier study and yet, no discussion of this earlier study is even mentioned as a rationale for the study.

---

## Round 0.2 · Minor Revisions

Please respond to the comments from Reviewer 2

·

Basic reporting

No Comment

Experimental design

This is a much-improved experimental design description, but in lines 138-139, is it 3 total measurements or 3 ROIs on x different 2D slices for a total of n measurements? Please clarify.

Validity of the findings

No comment

Additional comments

The paper is much improved and I thank the authors for their efforts. The introduction and discussion, my main concerns previously, now flows well and explains the reasoning behind this study. There are only two minor corrections the authors should make prior to publication, the clarification of the experimental design noted previously and in lines 26-27, 44-45, 96-97, and 108-109, it is sometimes stated as coastal vs. non-coastal and others it is near and not near the sea. Please be consistent.

---

## Round 0.3 · accepted · Accept

Congratulations!
Yours,
Yoshi
Prof. Yoshinori Marunaka, M.D., Ph.D.